# Proteomes of Residual Tumors in Curcumin-Treated Rats Reveal Changes in Microenvironment/Malignant Cell Crosstalk in a Highly Invasive Model of Mesothelioma

**DOI:** 10.3390/ijms232213732

**Published:** 2022-11-08

**Authors:** Daniel L. Pouliquen, Marine Malloci, Alice Boissard, Cécile Henry, Catherine Guette

**Affiliations:** 1Université d’Angers, Inserm, CNRS, Nantes Université, CRCI2NA, F-49000 Angers, France; 2Nantes Université, CHU Nantes, CNRS, Inserm, BioCore, US16, SFR Bonamy, F-44000 Nantes, France; 3Université d’Angers, ICO, Inserm, CNRS, Nantes Université, CRCI2NA, F-49000 Angers, France

**Keywords:** curcumin, tumor microenvironment, immune response, proteomics, biomarkers, T cells, malignant mesothelioma

## Abstract

Curcumin exhibits both immunomodulatory properties and anticarcinogenic effects which have been investigated in different experimental tumor models and cancer types. Its interactions with multiple signaling pathways have been documented through proteomic studies on malignant cells in culture; however, in vivo approaches are scarce. In this study, we used a rat model of highly invasive peritoneal mesothelioma to analyze the residual tumor proteomes of curcumin-treated rats in comparison with untreated tumor-bearing rats (G1) and provide insights into the modifications in the tumor microenvironment/malignant cell crosstalk. The cross-comparing analyses of the histological sections of residual tumors from two groups of rats given curcumin twice on days 21 and 26 after the tumor challenge (G2) or four times on days 7, 9, 11 and 14 (G3), in comparison with G1, identified a common increase in caveolin-1 which linked with significant abundance changes affecting 115 other proteins. The comparison of G3 vs. G2 revealed additional features for 65 main proteins, including an increase in histidine-rich glycoprotein and highly significant abundance changes for 22 other proteins regulating the tumor microenvironment, linked with the presence of numerous activated T cells. These results highlight new features in the multiple actions of curcumin on tumor microenvironment components and cancer cell invasiveness.

## 1. Introduction

Since the pioneering studies which started two decades ago, a plethora of evidence has been accumulated on the in vivo anticancer efficacy of curcumin and analogs in preclinical studies on various experimental tumor models [1]. In addition to a wealth of studies that document the fact that turmeric (*Curcuma longa* rhizome dried powder)/curcumin given in the diet prevents tumor formation or inhibits the progression of different types of cancer [2,3], curcumin administered intraperitoneally (i.p.) also inhibits the proliferation of human invasive cancer models [4,5]. Moreover, using an orthotopic model of peritoneal malignant mesothelioma (MM) established in immunocompetent rats, the repeated administration of curcumin given i.p. reduced tumor development, while an immune response was produced against residual tumor cells [6]. 

Bose et al. have previously reviewed the favorable interactions of curcumin with the immune system, including the restoration of CD4+/CD8+ T cell populations, the reduction in Treg cell populations and the reversion of type-2 cytokine bias [7]. Over the last five years, investigations on both experimental models and tumors from patients, using either injected or orally given curcumin, have provided considerable insights into the effects of curcumin on T cells linked with other immune cell infiltrating tumors. For example, in the latter case, curcumin was shown to regulate T helper 1 cells in patients with colon cancer [8], while converting Tregs from patients with lung cancer to T helper 1 cells [9]. In the former case, Liao et al. showed that the increase in CD8+ T cells produced by curcumin given i.p. was accompanied by a parallel decrease in Tregs and MDSCs [10]. Liu and colleagues revealed that a decrease in the expression of PD-1 and its ligands was associated with both the same increase in T cell proliferation and regulation of the epithelial-to-mesenchymal transition [11]. Finally, Hayakawa et al. observed that the augmentation of the in vivo induction of tumor antigen-specific T cells produced by curcumin was related to the restoration of dendritic cells through the inhibition of STAT3 [12].

We previously reported the presence of clusters of CD8+ T cells infiltrating the residual tumors collected in tumor-bearing rats treated with repeated injections of curcumin given i.p. [6]. Such cells were also observed in increased numbers in the liver of these curcumin-treated rats in comparison with untreated tumor-bearing rats [13]. Proteomic-based approaches have contributed to deciphering some of the complex mechanisms of the antitumor effects produced by curcumin and its derivatives. Although they were initially mostly restricted to in vitro studies on cultured cells [14,15,16], in vivo investigations have, for example, started to identify molecular targets of the liver colonization process by highly invasive MM cells [13]. Thus, this time we aimed to explore at the tumor level which changes in the microenvironment/malignant cell crosstalk were associated with the previously identified reduced tumor development and antitumor immune response produced by curcumin treatment [6]. The results show caveolin-1 represented a key player, linked with 115 other proteins, many of them involved in immune functions or signal transduction, cytoskeleton, metabolism, intracellular trafficking, nuclear, membrane or organelle processes.

## 2. Results

### 2.1. Tumors from Curcumin-Treated Rats Were Infiltrated with Numerous Activated Lymphocytes

Mesothelioma M5-T1 tumors collected from the three groups of rats (G1, untreated; G2, treated with curcumin on days 21 and 26; and G3, treated with curcumin four times on days 7, 9, 11 and 14) differed in several main histological features. In contrast with G1 tumors (see Appendix A and Section 4.2. for complementary information), the G2 and G3 tumors exhibited a decrease in cell density. Moreover, the G2 and G3 tumors were specifically characterized by large areas of necrosis and signs of fibrosis, respectively (Figure 1(2A,3B)). The sarcomatoid morphology of the M5-T1 tumor cells observed in G1 (Figure 1(1A,B)) also changed in the G2 and G3 tumors, especially in the inner parts of the tumor tissues (Figure 1(2B,3B)). The mitotic figures and isolated large macrophages found close to dividing cells, which were frequently observed in G1 (Figure 1(1C,D)), tended to disappear in the G2 and G3 groups. Finally, the G2 and G3 tumor tissues were characterized by the presence of numerous lymphocytes (Figure 1(2C,3C)), with their morphology progressively changing from a round (Figure 1(1E–G)) to a more fusiform shape (Figure 1(2E,F and 3D,F)), attesting their increased motility. Some of these lymphocytes were also frequently found associated with numerous monocytes/macrophages (Figure 1(2D)) close to necrotic areas (G2) or concentrated into the lumen of small vessels at the periphery of the G3 tumors (Figure 1(3E)). 

### 2.2. Residual Tumors from G2–G3 Curcumin-Treated Rats Exhibited Common Changes in 120 Proteins

All tissues were analyzed using the data independent acquisition (DIA) proteomic approach. The number of proteins detected in the two comparisons G2 vs. G1 and G3 vs. G1 were 3093 and 3096, respectively. In the first case, 693 proteins exhibited significant (*p* < 0.05) abundance changes, with 290 showing an increase and 403 a decrease. In the G3 vs. G1 comparison, this number increased to 833, with 514 showing an increase and 319 a decrease. Cross-comparing the two files, G2 vs. G1 and G3 vs. G1, led to a list of 116 common proteins, with 50 exhibiting an increase and 66 a decrease. The main location of these proteins recorded on https://www.proteinatlas.org (accessed on 5 July 2022) are illustrated in Figure 2, and their full names given in Appendix A. Among the proteins showing an increase, the most important ones already documented in the literature (Table 1) included caveolin-1 (encoded by *Cav1*), D-2-hydroxyglutarate dehydrogenase (a mitochondrial protein encoded by *D2hgdh*), Delta(24)-sterol reductase (encoded by *Dhcr24*), mammalian ependymin-related protein 1 (encoded by *Epdr1*), filamin-c (encoded by *Flnc*), inosine-5′-monophosphate dehydrogenase 1 (encoded by *Impdh1*), M-phase phosphoprotein 8 (encoded by *Mphosph8*), 5′-3′ exonuclease PLD3 (encoded by *Pld3*) and retinol-binding protein 4 (encoded by *Rbp4*) (Figure 3A). For proteins exhibiting a decrease, the most documented included beta-2-microglobulin (encoded by *B2m*), bone marrow stromal antigen 2 (encoded by *Bst2*), ectonucleoside triphosphate diphosphohydrolase I (encoded by *Entpd1*), Ena/VASP-like protein (encoded by *Evl*), immunity-related GTPase family M protein (encoded by *Irgm*), cytosol aminopeptidase (encoded by *Lap3*), multiple coagulation factor deficiency protein 2 homolog (encoded by *Mcfd2*), N-acylethanolamine-hydrolyzing acidamidase (encoded by *Nampt*), serine/threonine-protein kinase PAK1 (encoded by *Pak1*), SH3 domain-binding protein 1 (encoded by *Sh3bp1*), translocon-associated protein subunit gamma (encoded by *Ssr3*) and vascular cell adhesion protein 1 (encoded by *Vcam1*) (Figure 3B). 

### 2.3. Multiple and Early Curcumin Treatments Induced Specific Additional Events

We next examined which proteins presented a significant change in abundance between the rats given curcumin four times on days 7, 9, 11 and 14 (G3) versus those receiving curcumin twice on days 21 and 26 after the tumor challenge (G2). The aim was to identify the most important and simultaneous events associated with the observation of a high number of activated lymphocytes infiltrating the residual tumors in G3, as shown in Figure 1. We decided to focus on changes observed both in the comparisons of G3/G1 and G3/G2, and those found not to be significant in G2/G1. In association with the lymphocyte biomarkers, we first noticed a significant decrease in *Cd4* in both situations and a parallel tendency of an increase in *Cd8a* in G3/G2 (Figure 4 and Figure 5, top left), meaning a dramatic decrease in the *Cd4* to *Cd8a* ratio. Among all the proteins exhibiting increased or decreased abundance, the first selection, based on the observation of *p* < 0.0001 for at least one of the two comparisons of G3/G1 and G3/G2, led to a list of 65 proteins (with 45 showing an increase and 20 a decrease). The main locations of these proteins recorded on https://www.proteinatlas.org (accessed on 22 July 2022) are illustrated in Figure 4, and their full names are given in Appendix A. Interestingly, this list included 24 proteins which are completely or partly related to the modifications in the tumor microenvironment (Figure 4). In the second step, particular attention was given to those already documented in the oncology literature recorded in PubMed, and the evolution of their abundance changes for G2/G1, G3/G1 and G3/G2 is shown in Figure 5. 

Among the proteins involved in the immune function within the tumor microenvironment (Figure 5A), five exhibited an increase, including histidine-rich glycoprotein (encoded by *Hrg*), interleukin-33 (encoded by *Il33*), CD276 antigen (encoded by *Cd276*), leucine-rich repeat flightless-interacting protein 1 (encoded by *Lrrfip1*) and pro-low-density lipoprotein receptor-related protein 1 (encoded by *Lrp1*). Four proteins exhibited a decrease, including intercellular adhesion molecule 1 (encoded by *Icam1*), lysozyme C-1 (encoded by *Lyz1*), coronin-1A (encoded by *Coro1a*) and receptor-type tyrosine-protein phosphatase C (encoded by *Ptprc*). Three additional proteins, interferon regulatory factor 2 binding protein like (encoded by *Irf2bpl*), hexokinase-3 (encoded by *Hk3*) and interferon-induced transmembrane protein 3 (encoded by *Ifitm3*) appeared to be involved in the immune functions and signal transduction, metabolism and intracellular trafficking, respectively. Finally, the general transcription factor II-I (encoded by *Gtf2i*) represented another protein of interest involved in part of the regulation of angiogenesis (Figure 5A, fourth row). 

Another category corresponding to proteins playing a role in intracellular trafficking included caveolae associated protein 1 (encoded by *Cavin1*), microtubule-associated proteins 1A/1B light chain 3B (encoded by *Map1lc3b*) and protein OS-9 (encoded by *Os9*), which all increased (Figure 5B). The two proteins also involved in metabolic functions which presented an increase were glucose-6-phosphate isomerase (encoded by *Gpi*) and N(4)-(beta-N-acetylglucosaminyl)-L-asparaginase (encoded by *Aga*) (Figure 5C). Another two proteins of interest that increased were junction plakoglobin (encoded by *Jup*) and protein Niban 2 (encoded by *Niban2*), playing a crucial role in cell–cell interactions and apoptosis regulation, respectively (Figure 5D,E). With respect to signal transduction, four proteins were increased: tyrosine-protein phosphatase non-receptor type 11 (encoded by *Ptpn11*), mitogen-activated protein kinase 3 (encoded by *Mapk3*), glutamate receptor-interacting protein 1 (encoded by *Grip1*) and matrix remodeling-associated protein 8 (encoded by *Mxra8*); and one decreased, guanine nucleotide-binding protein G(i) subunit alpha-2 (encoded by *Gnai2*) (Figure 5F). Finally, one last category represented by four cytoskeletal proteins also increased: synaptopodin-2 (encoded by *Synpo2*), gelsolin (encoded by *Gsn*), dynamin-3 (encoded by *Dnm3*) and pleckstrin homology-like domain family B member 1 (encoded by *Phldb1*) (Figure 5G).

## 3. Discussion

The molecular mechanisms of action of curcumin against malignant tumor tissues are relevant to polypharmacology, a concept involving simultaneous interactions with multiple targets [64]. To date, our understanding of its in vivo effects means we must take into consideration the complexity of the signaling network organization [65], both at the intracellular level and within components of the tumor microenvironment. This task represents a terrible challenge that could be overcome, in part, using experimental tumor models designed in immunocompetent animals and high-throughput proteomics. Moreover, comparing the effects produced by different drug administration rates in the same model using syngeneic animals may produce additional value, especially when curcumin treatment generates an immune response directed against tumor cells. Inducing this immune response in curcumin-treated rats was previously established in this experimental model, characterized by the presence of numerous CD8+ T lymphocytes infiltrating both the residual tumors and the liver [6]. The proteome alterations accounting for the antitumor and antimetastatic effects of curcumin in the livers of these treated rats were also investigated [13]. Finally, determining the implication of the secondary lymphoid organs in the induced immune response was analyzed through their proteomic changes [66]. Thus, in this study, using the same approach, we identified the first set of protein abundance changes associated with the therapeutic efficacy of curcumin in vivo. Moreover, comparing the tumor proteomes from curcumin-treated rats submitted to two different timeline treatment schemes identified another set of proteins of interest related to the deep infiltration of the tumor tissue by activated T cells. 

On the first list, caveolin-1 represents a multifunctional protein that orchestrates many changes both in the tumor microenvironment/malignant cell crosstalk and intracellular functions. Its increased abundance found in common in the two groups of curcumin-treated rats is consistent with our histological observations of the presence of activated T lymphocytes in the residual tumors and with reports from the literature. The role caveolin-1 plays in the immune response has been well documented through the demonstration of T cell activation [17] and the organization of the immune receptors in their plasma membrane [20]. This protein also regulates immune cell infiltration [21] and promotes dendritic cell maturation [19]. Besides its effects on the immune components of the tumor stroma, the molecular interplay between *Cav1* and cell metabolism has been established [24], with its high expression suppressing c-Myc-induced metabolic reprogramming with an impact on breast stem cancer cells [25]. Moreover, this protein participates both in the intracellular communication between organelles and the signaling between cells [26] while inhibiting the unfolded protein response linked to ER stress [27]. Finally, caveolin-1 is also an inhibitor of the TGF-β signaling pathway, with an impact on the crosstalk between epithelial cancer cells and cancer-associated fibroblasts [22]. All these findings contribute to explaining the crucial role of stromal *Cav1* as a regulator of cytokine production and inflammation [18] and their harmful consequences on the poor clinical outcome observed in patients with decreased *Cav1* expression in breast [18,22], ovarian [23,28] and lung cancer [29]. 

In the first list, abundance changes to other proteins involved in immune functions were observed. These included, for example, the increase in two proteins encoded by the *Impdh1* and *D2hgdh* genes. In the first case, the highly proliferative state induced by T cell receptor engagement requires increased activity in the enzyme encoded by *Impdh1* [36]. In the second case, a recent work has reported the involvement of the enzyme encoded by this gene, which converts the oncometabolite D-2-hydroxyglutarate into 2-oxoglutarate in the enhancement of the antitumor effects of CAR-T cells in an immunosuppressive context [30]. In our study, the elevation of the activity of these two enzymes was thus consistent with the *Cav1* findings and our histological observations. Another line of evidence concerns the parallel decrease in the two proteins encoded by *B2m* and *Entpd1*. For example, several works have been reviewed to present a link between increased NK cell infiltration or activation and B2m-deficient tumors [45]. The increased level of the enzyme encoded by *Entpd1* (also known as CD39), which catalyzes the rate-limiting step of the conversion of extracellular ATP to adenosine in the tumor microenvironment, has been associated with metastasis [50]. A recent work has confirmed that CD39+ regulatory T cells (Tregs) were strongly involved in immunosuppression through the HIF-1-induced-upregulation of CD39 in hepatocellular carcinoma cells, which generated extracellular adenosine (eADO) stimulating the accumulation of immunosuppressive plasmacytoid dendritic cells [49]. Additionally, the isolation of tissue-infiltrating cells from a collection of tumors from colorectal cancer patients revealed a higher accumulation of CD39+ Tregs promoting tumor progression and metastasis, leading to poor diagnosis [51]. Accordingly, two other works have reported that a PD-1 blockade restored the helper activity of PD-1^hi^CD39+ CD4 T cells presenting an exhausted state [47], while Entpd1 expression affected T cell receptor diversity; two additional observations which tended to confirm its critical role in T cell exhaustion [48]. 

Interestingly, the significant amplification of abundance change observed between G3 vs. G1, relative to G2 vs. G1, which was observed for *B2m* and *Entpd1*, was shared by eight proteins of interest involved in other functions. Of these, three proteins encoded by the *Flnc*, *Pld3* and *Rbp4* genes were concerned with an increase, while five others, encoded by the *Bst2*, *Irgm*, *Nampt*, *Sh3bp1* and *Vcam1* genes, were concerned with a decrease. *Flnc* was reported to represent one of seven genes linked to a biochemical relapse-free survival in patients with prostate cancer [34]. Filamin-C belongs to a family of three actin-binding proteins that stabilize actin networks and connect them to the cell membrane, with its silencing associated with cancer-increased invasiveness through the lamellipodia formation [35]. *Pld3* represents an immune-related gene, with its low expression leading to a lower survival in patients with osteosarcoma [39]. This exonuclease (also named phospholipase D3) was recently found to be transcriptionally regulated by p53, with its higher expression representing a good prognostic factor in patients with pancreatic cancer [40]. The downregulation of this enzyme is also an adaptation of lipid metabolism in hypoxic cancer cells, which is mediated by HIF-1 [38]. Regarding retinol-binding protein 4, the low expression of Rbp4 has previously been associated with a poorer prognosis in patients with renal cell [41] and hepatocellular carcinomas [42] or breast cancer [43]. Interestingly, a recent single-cell sequencing study analyzing the differences between right- and left-sided colorectal cancer tissues revealed that the specific presence of a subpopulation of RBP4+ cancer cells in the latter was associated with a higher ratio of pre-exhausted/exhausted T cells and an increased responsiveness to immunotherapy [44]. In our study, the continuous decrease observed in the levels of Bst2 and Irgm was also consistent with reports describing their role as survival promoters in breast cancer [46] and melanoma cells [55]. Tumor invasion was also described as being promoted by an increased expression of *Sh3bp1* and *Vcam1* in hepatocellular carcinoma [61] and gastric cancer [63], respectively. Finally, our observation of decreased levels of NAMPT in curcumin-treated rats are in good agreement with the role of this protein in stemness activation [58,60].

For another set of proteins, no significant differences were observed in the comparison of the two groups of curcumin-treated rats; however, two of them exhibited one of the highest and lowest fold changes in G2 vs. G1, respectively, providing additional meaning, at the intracellular level, to the modifications to the tumor microenvironment discussed above. The first case concerned a nuclear protein member of the human silencing hub (HUSH) complex involved in epigenetic repression, encoded by *Mphosph8* and frequently downregulated in diverse cancers, leading to DNA damage [37]. The second case corresponded to an ER-Golgi protein, encoded by *Mcfd2*, that plays an important role in controlling stem cell pluripotency and is involved in cancer progression and metastasis [57]. Interestingly, curcumin treatment affected another ER protein (SSRG, encoded by the *Ssr3* gene). This protein is part of the translocon complex, and its overexpression was associated with poor survival in patients with hepatocellular carcinoma [62]. In a previous work, we showed that the level of this protein involved in immune signaling was increased in the livers of rats invaded by M5-T1 tumor cells, with this increase being reversed by curcumin treatment [13]. This protein was listed in the most important correlations found between the 179 biomarkers of liver colonization identified in this study, together with the beta-2-microglobulin (B2MG, encoded by *B2m*) discussed above [13]. Thus, we conclude these two proteins represent two candidate biomarkers relating ER-Golgi to ECM events, the elevated levels of which are affected by curcumin treatment both at the tumor level and in a preordained microenvironment hospitable to invasive cancer cells, such as the liver. Other genes encoding proteins of interest include *Epdr1*, a transmembrane protein involved in cell–cell adhesion, which suggests that the morphological changes affecting M5-T1 tumor cells that we observed in the tumor tissues from curcumin-treated rats were associated with a decrease in their sarcomatoid character. Its overexpression in breast cancer tissues was reported to lead to an antitumorigenic effect mediated by the p53 signaling pathway [32]. The parallel decrease in PAK1 also attests to the reduced progression stage of the curcumin-treated tumors, as this effector of the Rho GTPases (encoded by *Pak1*), involved in cell motility and cytoskeleton remodeling, is overexpressed in almost all tumors [60]. Finally, one last line of evidence of this reduced invasiveness is given by the decrease in the Ena/VASP-like protein (encoded by *Evl*), which organizes stress fibers leading to cancer cell stiffening [52], with stiffness playing an important role in the harmful effects of epithelial-to-mesenchymal transition (EMT) in invasive cancers [67]. 

The second list of proteins, for which significant specific changes were observed both in the G3 vs. G1 and G3 vs. G2 groups, provides additional insights into the role of several proteins of interest involved in immune functions. The decrease in CD4, associated with a tendency towards an increase in CD8, led to a dramatic decrease in the CD4 to CD8 ratio, a result consistent with previous work demonstrating it represented a favorable prognosis marker for overall survival in vaccine-treated ovarian cancer patients [68]. Patients with gastric cancer and a low CD4 to CD8 ratio also exhibited 3.6-times higher overall survival compared with patients with a high ratio [69]. Linked with these findings, the potent cytokine that represents interleukin-33, specifically found increased in G3, adds further proof attesting to the dramatic changes that occurred in the microenvironment/cancer cell crosstalk, given its role in the recruitment/expansion of immune cells and the growth reduction in solid tumors [70]. Another interesting point was the increase in the protein encoded by *Lrrfip1*, related to the presence of multinucleated giant cells, formed by fusion of circulating monocytes, involved in the phagocytosis of cancer cells and potentially associated with better survival in esophageal cancer patients [71]. The implication of LRP1 (also known as CD91, encoded by *Lrp1*) in tumor immunosurveillance has been well documented, providing a highly efficient conduit for the cross-presentation of tumor antigens to T cells [72]. Among the proteins exhibiting a decrease, ICAM-1 (encoded by *Icam1*) represents an adhesive molecule also known to weaken the immune response in cancer cells in addition to its promoting action on cancer progression [73]. The protein tyrosine phosphatase CD45 (encoded by *Ptprc*) was shown to affect the function of T and other immune cells, with a negative impact on the prognosis of lung adenocarcinoma [74]. 

The histidine-rich glycoprotein (encoded by *Hrg*) is one of three proteins whose multiple functions include interactions with numerous ligands. The dramatic increase we specifically observed after multiple early curcumin treatments, in the context of the immune changes discussed above, suggests it is a cornerstone of many changes at the interface between cancer cells and the tumor microenvironment. First, the action of HRG on the polarization of tumor-associated macrophages (TAMs) from an immunosuppressive M2- to a tumor-inhibiting M1-like phenotype contributes to promoting antitumor immune responses [75]. Secondly, the HRG gene delivery on a mouse model of intracranial glioma resulted in an increased accumulation of CD8+ T cells in the tumor, thus contributing to leukocyte differentiation in parallel with reduced glioma growth [76]. Thirdly, regarding NK cells, HRG was shown to augment their killer function by modulating PD-1 expression [77]. Finally, in addition to its action on immune cells, HRG also affected endothelial cells, suppressing their adhesion, spreading and migration on collagen I, with these alterations leading to changes in the tumor vessel phenotype [78]. Additionally, at the intracellular level, two other proteins are likely to participate in the overall tumor changes. IRF2BPL is one of three members of a family of transcriptional regulators that potentially control various hallmarks of oncogenesis, including tumor suppression depending on the context [79]. A recent work has demonstrated that it downregulates Wnt signaling by interacting with and targeting β-catenin (crucial markers of EMT in cancer [67]) for proteasomal degradation [80]. The second protein, a member of the hexokinase family encoded by *Hk3*, participates in tumor immune microenvironment remodeling and promotes immune escape, with its high expression leading to a poor prognosis in patients with renal carcinoma [81]. 

Interestingly, apart from protein abundance changes relative to immune functions, additional events were observed. The first protein of interest was represented by synaptopodin-2 (encoded by *Synpo2*). In our study, the increased level of this protein in G3 agreed with two complementary findings reported by two independent teams four years ago, revealing its suppression of breast cancer metastasis via the inhibition of two signaling pathways, PI3K/Akt/mTOR [82] and YAP/TAZ [83]. Since then, the confirmation of the negative effects of SYNPO2 on migration and EMT was also established in colorectal cancer cells, through the suppression of hypoxia-induced progression [84]. Another line of evidence of loss in the sarcomatoid character of G3 residual tumor tissues was provided by the increase in junction plakoglobin (encoded by *Jup*). This result can be related to the observations made on its knockdown on gastric cancer cells, which caused EMT and promoted cell migration [85], and its modulating effect on HIF2 signaling during renal cell carcinoma progression [86]. Within candidate biomarkers involved in signal transduction, the decreased level of the protein encoded by *Gnai2* raises a question linked with a recent work showing its role in onco-transcriptome regulation, revealing 264 genes were dependent on its expression [87]. This gene was also listed in the most important correlations found between the 179 biomarkers of liver colonization identified in our previous study [13]. Thus, both the curcumin-induced reversion of the increase observed in the context of the liver metastatic colonization [13] and the decrease found in the residual tumor in this study attests to its value as a biomarker of therapeutic efficacy. Finally, the rise observed in the levels of two proteins encoded by *Gtf2i* and *Cavin1* suggests additional important information needs to be considered from the literature. The first information refers to the discovery of the antagonistic role of GTF2I (also known as TFII-I) against GATA2, with these two transcription factors governing the expression of the VEGF receptor [88]. The second information is the enhancement of prostate cancer cell migration and invasion that was observed in stromal cells lacking CAVIN1, leading to a poor outcome in patients [89]. 

## 4. Materials and Methods

### 4.1. Experimental In Vivo Experiments

F344 male Fisher rats, purchased from Charles River Laboratories (L’Arbresle, 69, France) were maintained under SPF (specific pathogen-free) status in the UTE-IRS UN Animal Facility at Nantes Université. Experiment protocols followed the European Union guidelines for the care and use of laboratory rodents in research protocols. The experiments were approved by the Ethics Committee for Animal Experiments (CEEA) of the Pays de la Loire Region of France (2011.38) and #01257.03 of the French Ministry of Higher Education and Research (MESR). The M5-T1 original neoplastic cell line (https://technology-offers.inserm-transfert.com/offer/rat-mesothelioma-cell-line-m5-t1/ (accessed on 19 August 2022)) [6] was injected intraperitoneally (3 *×* 10^6^ cells in 200 µL PBS buffer) into all rats on day 0. For group G1 (*n* = 2, selected from a group of 9 untreated tumor-bearing rats, see Appendix A and Section 4.2. for complementary information), the animals were anesthetized in an isoflurane chamber (Forene^®^, TemSega, Pessac 33600, France, Abbott, Rungis Cedex, France) on day 21 and euthanized in their home cage with a rate of 30% volume displacement per minute of CO_2_. Solid tumors growing on the omentum, metastases and associated nodules were collected and immediately fixed. In the G2 group (*n* = 3), rats were given two successive injections i.p. of 750 µg/kg of curcumin on days 21 and 26 [6]. The animals were anesthetized and euthanized, as described above, the day after, and softer tumors presenting diffuse yellowish staining were collected and fixed. In group G3 (*n* = 6), the animals were given 4 successive injections i.p. of 1.5 mg/kg of the same curcumin solution on days 7, 9, 11 and 14 [6]. The animals were anesthetized and euthanized, as described above, two days after (day 16), and the remaining tumors present in the peritoneal cavity were collected and fixed. For both histological and proteomic analyses, the small residual tumors (collected from *n* =3 rats) representing the subgroup showing a dramatic reduction in total tumor masses (previously analyzed in [6] and [13] for other purposes) were selected. The quality of the 10mM stock solution of curcumin (dissolved in DMSO) used for treatment was also checked by ^1^H and ^13^C NMR spectroscopy, as previously described [13].

### 4.2. Histological Analyses

Tumor tissue samples collected from the G1, G2 and G3 groups of rats and fixed in 4% paraformaldehyde (in PBS buffer) were embedded in paraffin and cut with a Leica RM2255 microtome (Leica Biosystems, Nussloch, Germany). For group G1, two of nine samples collected from nine tumor-bearing rats were selected based on the following criteria. Firstly, primary tumor nodules growing on the omentum were preferred to metastatic tumor masses (invading the liver, muscle tissue, the pancreas or the gut). The aim was to avoid the comparison of tumor samples from G2 and G3 groups with tumor samples from rats in an advanced stage of tumor progression within the G1 group [90]. This selection was justified by the fact that the two groups of M5-T1 tumors differed in the relative abundance of 148 proteins, as shown previously [90]. Secondly, samples **1** and **2** (Appendix A) were selected as they both presented dense and large tumor masses in good preservation state, allowing a careful morphological analysis of TILs at high resolution. Three-µm-thick sections of all samples were stained with hematoxylin phloxine saffron (HPS), and the slides were scanned (NanoZoomer 2.0 HT Hamamatsu, Japan) for preliminary checks of areas of interest. Tumor areas corresponding to previous histological examinations (immunohistochemical identification of CD3+ and CD8+ lymphocytes and lymphocyte quantification, as described in [6] and [13]), were selected and viewed using high magnification for analyses of morphological changes, as illustrated in Figure 1. 

### 4.3. Proteomic Analyses

For DIA analyses, four 20-µm-thick sections of each tumor tissue sample were used. Selected areas were removed with a scalpel and collected in a 1.5-mL Eppendorf^®^ microtube. The samples were deparaffinized with three successive xylene washes, rehydrated in decreasing-grade ethanol solutions and vacuum-dried [13]. Dried tissues were resuspended in 200 µL of Rapigest SF (Waters, Milford, MA, USA), and dithiothreitol was added to a final concentration of 5 mM (DTT, AppliChem, Darmstadt, Germany). Samples were incubated in a thermo shaker at 95 °C for one hour, and sonication performed twice using an ultrasonic processor 75,185 (130 W, 20 KHz, Bioblock Scientific, Illkirch, France). Subsequently, cysteine residues were alkylated by adding 200 mM S-Methyl methanethiosulfonate (MMTS) to a final concentration of 10 mM (incubated at 37 °C). Sequencing-grade trypsin was added in a ratio ≥2 µg mm^−3^ tissue (incubated at 37 °C overnight). The reaction was stopped with formic acid (9% final concentration) and incubated at 37 °C for one hour, and the acid-treated samples were centrifuged at 16,000× *g* for ten minutes. Salts were removed from the supernatant and collected in new reaction microtubes using self-packed C18 STAGE tips. Peptide concentrations were finally determined with the Micro BCA™ Protein Assay Kit (Thermo Fisher Scientific, Saint-Herblain, France).

Two hundred nanograms of each sample were analyzed by LC-MS/MS with a nanoHPLC (NanoElute, Bruker Daltonik GmbH, Bremen, Germany) coupled to a TimsTOF Pro2 (Bruker Daltonik GmbH, Bremen, Germany). Measurements were acquired in diaPASEF mode (data independent acquisition parallel accumulation serial fragmentation). Mass spectrometry data were analyzed using Spectronaut^TM^ v16.0 (Biognosys AG, Schlieren, Switzerland).

The direct DIA workflow in Spectronaut^TM^ was used for analyzing the dia-PASEF dataset with no need to build a library from the DDA runs (library-free) with default analysis settings. Retention time prediction type was set to dynamic iRT, and calibration mode was set to automatic. These settings also included mutated decoy method, and cross run normalization was enabled. The FDR (faults discovery rate) was estimated with the mProphet approach, and Q-value cut-off for both precursor and protein was set to 1%. Interference correction was enabled for quantification which required a minimum of 2 precursor ions and 3 fragment ions. The term protein refers to protein groups as determined by the algorithm implemented in Spectronaut^TM^. The statistical testing for differential abundance is performed by paired Student’s *t*-test using Spectronaut^TM^.

## 5. Conclusions

One of the main conclusions of our study is that the repeated administration of curcumin to tumor-bearing rats produces significant abundance changes in a set of proteins associated with the tumor microenvironment. In agreement with our previous findings in the same experimental model at the liver and lymphoid organ levels, the modifications observed in several immunological markers tend to confirm the hypothesis of an out-of-field systemic antitumor effect. Among the genes encoding abundant proteins located in the plasma and platelets that may orchestrate these changes, *Hrg* appears to play an important role. The associated protein, produced by hepatocytes, is involved in the regulation of the immune complex, cell chemotaxis and adhesion, angiogenesis and coagulation; thus, it represents a link between the tumor microenvironment and liver functions. Although a multifaceted relationship with cancer has been described for *Hrg*, other events provide additional lines of evidence of this link with the liver. The case of *Rbp4* is of special interest, as this gene encodes a protein that mediates retinol transport from the liver stores to the peripheral tissues. The gene *Nampt*, which encodes another protein secreted to blood and whose level was dramatically decreased in our study, represents a prognostic marker in all cancer types. The dramatic increased level of the HUSH complex member, encoded by *Mphosph8*, confirmed previous reports on the epigenetic modulation of target genes by curcumin, emphasizing the interest of drugs relevant to polypharmacology. The combined increased abundances observed in the proteins encoded by *Cav1*, *Pld3* and *Hk3* also highlight the interplay with tumor cell metabolism modifications, leading to the suppression of cancer cell stemness and the deleterious effects of cancer-associated fibroblasts. Finally, among the many other findings providing exciting prospects for future studies, the inter-relationships between tumor microenvironment changes and cytoskeletal events involving, for example, the *Flnc* and *Synpo2* genes raise numerous questions.

## Figures and Tables

**Figure 1 ijms-23-13732-f001:**
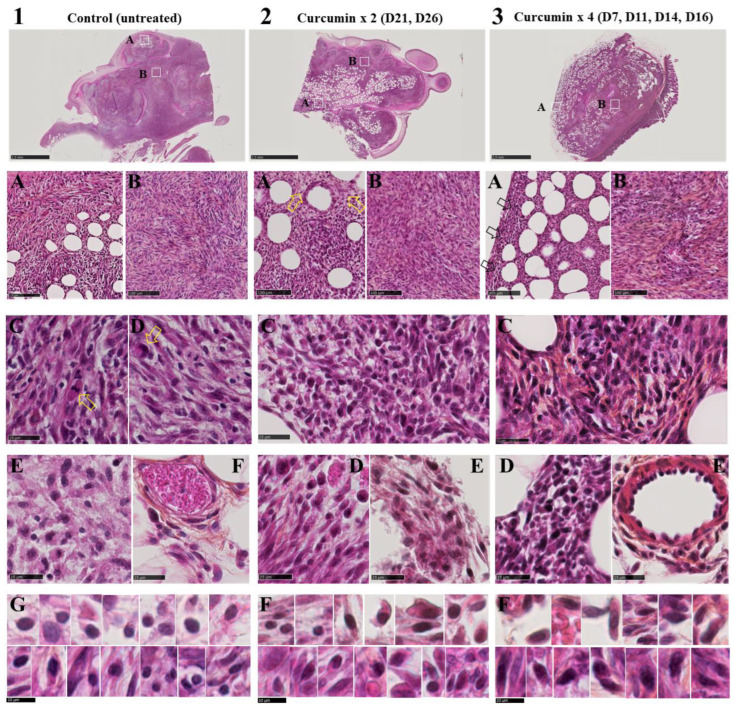
**Tumor histological features of M5-T1 mesothelioma tumors. Top, first row**: general views of representative tumors from the three groups of rats used for proteomic analysis. HPS staining rendering nuclei, cytoplasm and collagen, respectively, blue-violet, pink and yellow orange. The scale bars represent 2.5 mm. (**Second row**, **A**,**B**): magnifications of areas delimited by the white squares shown on general views. The scale bars represent 100 µm. Photograph (**2A**) shows a decrease in cancer cell density (open yellow arrows) in areas exhibiting necrosis produced by two successive injections of curcumin on days 21 and 26 after tumor challenge. Photograph (**2B**) illustrates morphological changes of cancer cells which contrasted with the typical sarcomatoid morphology observed in both external and internal views of the tumors from control (untreated) rats (**1A,B**). Photograph (**3A**) shows massive infiltration of this residual tumor by lymphocytes (open black arrows) after four successive injections of curcumin on days 7, 11, 14 and 16 after tumor challenge. The internal part was also characterized by morphological changes of cancer cells (as in **2B**) and important signs of fibrosis (**3B**). **Third row**: high magnification views of each tumor type (the scale bars represent 25 µm) showing the presence of many mitotic figures (**1C**) and isolated large macrophages (**1D**) (open yellow arrows) close to cancer cells exhibiting sarcomatoid morphology. In contrast, cancer cells from rats treated twice or with four successive injections of curcumin revealed significant morphological changes, with the presence of numerous immune cells (**2C**,**3C**), sometimes associated with fibrosis (**3C**). **Fourth row**: high magnification views of the external part of the tumors, showing rare spherical lymphocytes in untreated rats (**1E**,**F**) while starting to change their morphology (**2D**,**E**) and finally, spreading to all parts of the tumor (**3D**). Note their presence in high numbers (**3E**), with some of them crossing the tunica media of the post capillary venule (which contrasted with the situation illustrated in (**1F**) for untreated rats). **Fifth row**: morphological changes of tumor-infiltrating lymphocytes (TILs) associated with the development of an immune response directed against M5-T1 tumor cells in curcumin-treated rats. High magnification views from 6–7 different fields from the external (top row) and internal parts (bottom row) of tumors from the three groups of rats (scale bars represent 10 µm). A progressive evolution towards an increase in the cytoplasmic to nucleus ratio/amoeboid migration (associated with cytoplasmic protrusions and elongated morphology) was observed from (**1G**) to (**2F**) and (**3F**).

**Figure 2 ijms-23-13732-f002:**
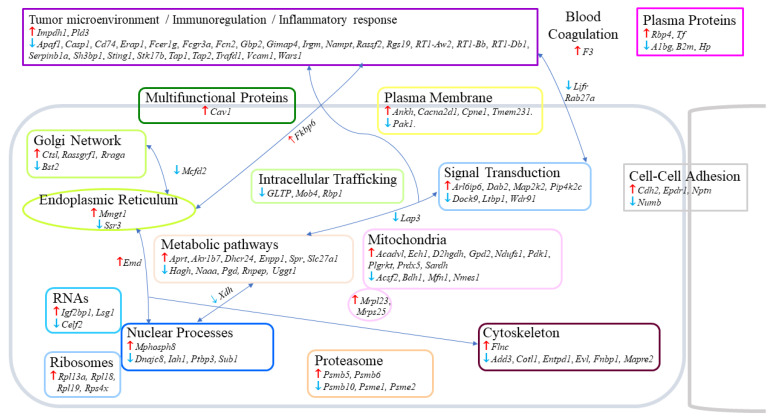
**Changes observed in the proteome of the M5-T1 tumor following treatment with curcumin.** Proteins are identified by their encoding genes (*in italics*), with **↑** and **↓** symbolizing significant common increase or decrease relative to tumors from untreated rats, respectively. The list of 116 proteins (full names given in Appendix A) was established by crossing the two lists G2 vs. G1 and G3 vs. G1 (satisfying the condition *p* < 0.05).

**Figure 3 ijms-23-13732-f003:**
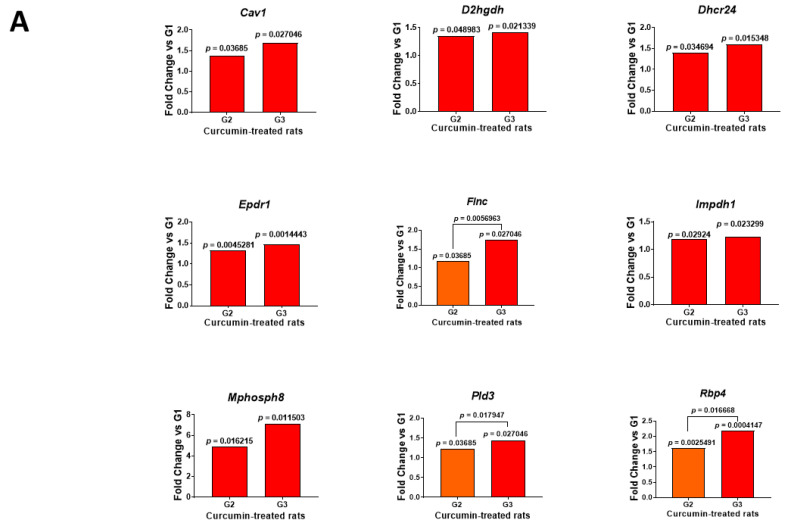
Evolution of abundance changes in the main proteins already documented in the literature. (**A**): increase. (**B**): decrease. The results are expressed as fold changes for G2 or G3 vs. G1, with *p* values observed for G2 vs. G1 and/or G3 vs. G1 indicated at the top of the bars. When the differences between G3 vs. G2 were also significant, corresponding *p* values were also included.

**Figure 4 ijms-23-13732-f004:**
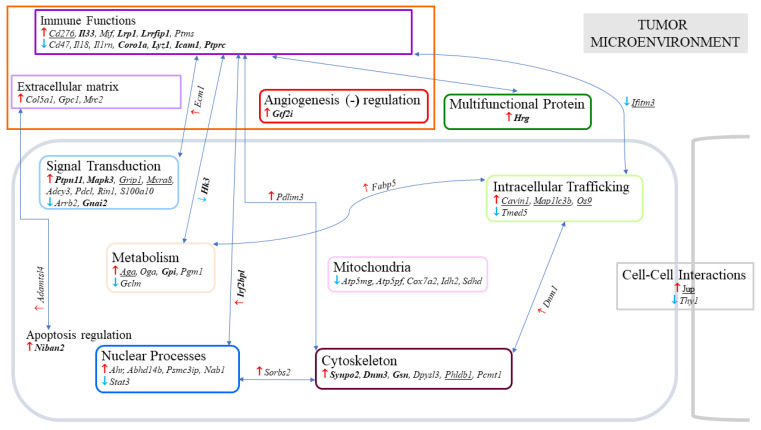
Main proteins exhibiting specific abundance changes related to multiple curcumin treatments. Proteins are identified by their encoding genes (*in italics*), with **↑** and **↓** symbolizing significant increase or decrease found in common in G3 vs. G1 and G3 vs. G2 and not found significant in G2 vs. G1. Underlined gene names correspond to *p* < 0.001 for the two conditions G3 vs. G1 and G3 vs. G2. Gene encoding proteins for which the evolution of abundance changes are shown in Figure 5 and are indicated in bold. Arrows between boxes symbolize proteins having multiple locations, as recorded on the website https://www.proteinatlas.org (accessed on 22 July 2022).

**Figure 5 ijms-23-13732-f005:**
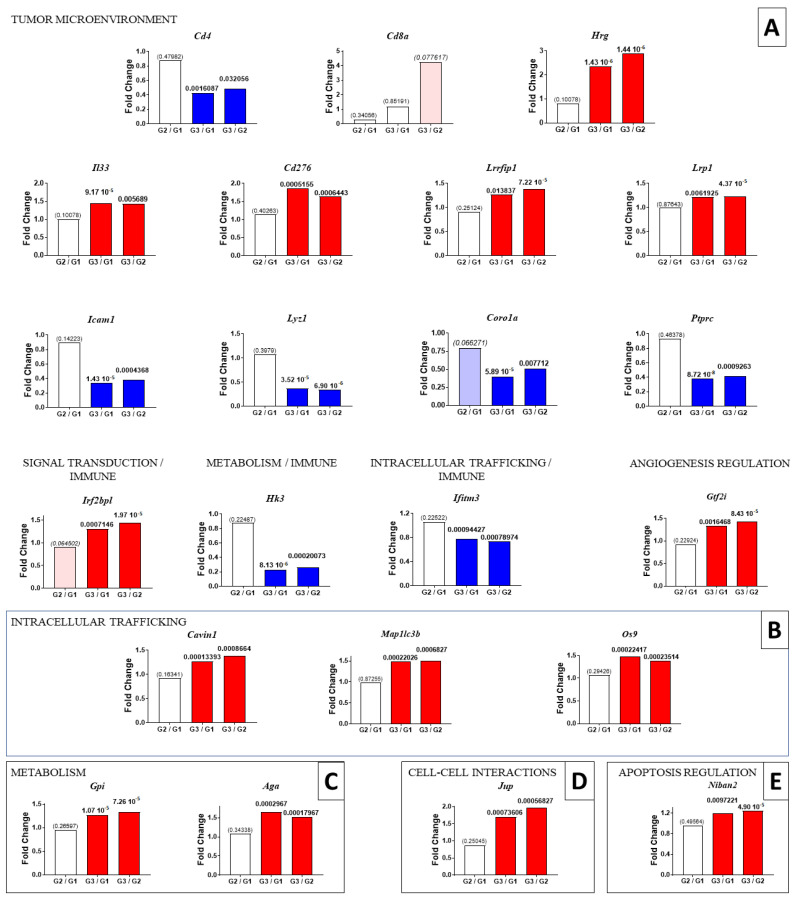
**Specific abundance changes related to multiple early curcumin treatments.** (**A**): proteins involved in tumor microenvironment. (**B**): proteins involved in intracellular trafficking. (**C**): cell metabolism enzymes. (**D**): proteins involved in cell-cell interactions. (**E**): proteins involved in apoptosis regulation. (**F**): proteins involved in signal transduction. (**G**): cytoskeletal proteins. The results are shown for a selection of proteins based on the most significant *p* values observed in the comparison of G3/G1 and G3/G2 (indicated in bold at the top of the bars). Proteins are identified by their encoding genes (*in italics*). For nonsignificant differences observed in the comparison of G2/G1, *p* values are indicated in brackets and for tendencies in italics (in brackets).

**Table 1 ijms-23-13732-t001:** **Main proteins exhibiting common changes in residual tumors from the G2 and G3 groups of curcumin-treated rats (relative to G1 untreated rats)**. Proteins are identified by their encoding genes (*in italics*), with full names (as recorded on https://www.uniprot.org (accessed on 22 September 2022) for rattus norvegicus) given in the text. The list provided was selected from the 116 proteins appearing in Figure 2, based on their documented role reported in the literature (PubMed) in recent years.

Gene	Authors (Year)	Main Findings
*Cav1*	Ohnuma et al., 2007 [17]	Triggers T cell activation
	Martinez-Outschoorn et al., 2011 [18]	Loss of stromal Cav1 predictor of poor outcome
	Oyarce et al., 2017 [19]	Increases maturation of dendritic cells
	Schaffer and Minguet, 2020 [20]	Lymphocyte cell membrane organization
	Song et al., 2022 [21]	Regulates immune cell infiltration
	Pavlides et al., 2009 [22]	Vimentin overexpression in tumor lacking Cav1
	Saeed-Vafa et al., 2021 [23]	Loss of Cav1 in malignant ovarian carcinomas
	Nwosu et al., 2016 [24]	Cav1 modulates cell metabolism
	Wang S. et al., 2020 [25]	Inhibits breast cancer stem cells
	Simon et al., 2020 [26]	Regulates intracellular organelle communication
	Diaz et al., 2020 [27]	Inhibition of the unfolded protein response
	Yang et al., 2021 [28]	Exosomal Cav1 downregulated in ovarian cancer
	Yin et al., 2022 [29]	Worse survival in low Cav1 lung cancer patients
*D2hgdh*	Yang et al., 2022 [30]	Overexpression increases CAR-T cell killing efficacy
*Dhcr24*	Zerenturk et al., 2013 [31]	Involved in cholesterol biosynthesis
*Epdr1*	Liang et al., 2020 [32]	Increases p53, p21 and Bcl-2 expression
	Zhao et al., 2021 [33]	Tumor suppressor (PI3K/Akt signaling pathway)
*Flnc*	Aakula et al., 2016 [34]	Increases free survival in prostate cancer patients
	Kokate et al., 2018 [35]	Downregulation increases gastric cancer invasiveness
*Impdh1*	Duong-Ly et al., 2018 [36]	Upregulated in T cell activation
*Mphosph8*	Tunbak et al., 2020 [37]	Frequently downregulated in diverse cancers
*Pld3*	Valli et al., 2014 [38]	Reduced PLD3 levels in hypoxia
	Guo et al., 2021 [39]	Better prognosis in high PLD3 osteosarcoma
	Butera et al., 2022 [40]	Correlates with p53 status and prognosis
*Rbp4*	Sobotka et al., 2017 [41]	Poorer prognosis of renal carcinoma, low RBP4
	Li M. et al., 2021 [42]	Poor prognosis of liver cancer with low RBP4
	Wu et al., 2021 [43]	Low expression level in breast cancer tissues
	Guo et al. 2022 [44]	Link with responsiveness to immunotherapy
*B2m*	Wang H. et al., 2021 [45]	B2M-deficient tumors express activating NK ligands
*Bst2*	Mahauad-Fernandez et al., 2018 [46]	Promotes survival in metastatic breast cancer
*Entpd1*	Balança et al., 2021 [47]	Involved in tumor-infiltrating exhausted T cells
	Kim et al., 2021 [48]	Entpd1-T cells have increased TCR diversities
	Pang et al., 2021 [49]	HIF-1 upregulates Entpd1 in liver cancer
	Whitley et al., 2021 [50]	Entpd1 expression linked with metastasis
	Zhan et al., 2021 [51]	Entpd1+ Tregs correlates with poor prognosis
*Evl*	Tavares et al., 2017 [52]	Regulates F-actin and promotes cell stiffening
	Chen et al., 2018 [53]	Upregulated in gastric cancer, link with stage
*Irgm*	Xu et al., 2019 [54]	Upregulated in glioma, link with autophagy
	Tian et al., 2020 [55]	Promotes melanoma cell survival via autophagy
*Lap3*	He et al., 2015 [56]	Promotes glioma progression and invasion
*Mcfd2*	Fukamachi et al., 2018 [57]	Crucial component in oral cancer metastasis
*Nampt*	Navas and Carnero, 2021 [58]	Activates stemness and dedifferentiation
	Nacarelli et al., 2022 [59]	NAMPT inhibition suppresses cancer stem-like cells
*Pak1*	Kumar and Li, 2016 [60]	Converging nexus for cancer-promoting signals
*Sh3bp1*	Tao et al., 2016 [61]	Promotes tumor invasion/microvessel formation
*Ssr3*	Huang et al., 2018 [62]	Predicts poor survival in patients with liver cancer
*Vcam1*	Shen et al., 2020 [63]	Facilitates tumor invasion in gastric cancer

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
