# Peer review of "Proteomes of Residual Tumors in Curcumin-Treated Rats Reveal Changes in Microenvironment/Malignant Cell Crosstalk in a Highly Invasive Model of Mesothelioma"

_ijms, 2022, doi:10.3390/ijms232213732_

Round 1
Reviewer 1 Report
In the manuscript entitled Proteomes of residual tumors in curcumin-treated rats reveal changes in microenvironment / malignant cell crosstalk in a highly invasive model of mesotheliomathe, authors show study in which they used a rat model of highly invasive peritoneal mesothelioma to analyze residual tumor proteomes of curcumin-treated rats, in comparison with untreated tumor-bearing rats, to provide insights into modifications in tumor microenvironment malignant cell crosstalk.
I notice that the number of animals in the experimental groups is very small (n=2 or n=3). That is a small number for statistical analysis. This part needs to be improved.
Reviewer 2 Report
I read with great interest the article " Proteomes of residual tumors in curcumin-treated rats reveal changes in microenvironment / malignant cell crosstalk in a highly invasive model of mesothelioma” by Daniel L Pouliquen , Marine Malloci , Alice Boissard , Cécile Henry and Catherine Guette.
In my opinion, the article is well-written, structured and the material is well-chosen. Results correctly presented and extensively visualized. Discussion was conducted well.
In this case, curcumin was used - natural bioactives with anti-carcinogenic properties in a highly invasive model of mesothelioma.
In terms of research, I rate the work very highly. I believe that the work is fully understandable and does not require any corrections.
Reviewer 3 Report
In this work, the authors explore the multiple effects of curcumin on tumor microenvironment and malignant cell crosstalk from proteomics level by comparison analysis . And they found that several important proteins that participated in regulation of tumor microeviroment. Especially the caveolin-1 which represented a key player and linked with 115 other proteins, many of them involved in immune functions, intracellular trafficking, membrane or organelle processes and so on. The paper addresses an interesting and timely topic and presents interesting findings. However, the major deficiency is the proteomics analysis. The main analysis methods and used statistics methods should be clearly stated. For example, which test was used? Hence, please add one or two paragraphs in the methods part to state the analysis methods and corresponding parameters.
Round 2
Reviewer 1 Report
Thank you for accepting my request and improving your manuscript
Reviewer 3 Report
The authors addressed my comments. There is no further comment.